# Palaeogenomic analysis of black rat (*Rattus rattus*) reveals multiple European introductions associated with human economic history

He Yu[1,2,3,61], Alexandra Jamieson[4,61], Ardern Hulme-Beaman[5,6], Chris J. Conroy [7], Becky Knight[8], Camilla Speller[9,10], Hiba Al-Jarah[9], Heidi Eager[11], Alexandra Trinks[4,12], Gamini Adikari[13], Henriette Baron[14], Beate Böhlendorf-Arslan[15], Wijerathne Bohingamuwa [16], Alison Crowther[17,18], Thomas Cucchi[19], Kinie Esser[20], Jeffrey Fleisher [21], Louisa Gidney[22], Elena Gladilina[23], Pavel Gol'din [24], Steven M. Goodman [25], Sheila Hamilton-Dyer[26], Richard Helm [27], Jesse C. Hillman[28], Nabil Kallala[29,30], Hanna Kivikero[31,32], Zsófia E. Kovács[33], Günther Karl Kunst[34], René Kyselý [35], Anna Linderholm [4,36], Bouthéina Maraoui-Telmini[37], Nemanja Marković [38], Arturo Morales-Muñiz[39], Mariana Nabais [40,41], Terry O'Connor[9], Tarek Oueslati [42], Eréndira M. Quintana Morales [43], Kerstin Pasda[44], Jude Perera[45], Nimal Perera[45], Silvia Radbauer[46], Joan Ramon[47], Eve Rannamäe [48], Joan Sanmartí Grego[49], Edward Treasure[50], Silvia Valenzuela-Lamas [51], Inge van der Jagt[52], Wim Van Neer [53,54], Jean-Denis Vigne[19], Thomas Walker[55], Stephanie Wynne-Jones [8], Jørn Zeiler[56], Keith Dobney [5,57,58,59], Nicole Boivin [17,18], Jeremy B. Searle [11], Ben Krause-Kyora [60], Johannes Krause [1,3✉], Greger Larson [4✉] & David Orton [9✉]

The distribution of the black rat (*Rattus rattus*) has been heavily influenced by its association with humans. The dispersal history of this non-native commensal rodent across Europe, however, remains poorly understood, and different introductions may have occurred during the Roman and medieval periods. Here, in order to reconstruct the population history of European black rats, we first generate a de novo genome assembly of the black rat. We then sequence 67 ancient and three modern black rat mitogenomes, and 36 ancient and three modern nuclear genomes from archaeological sites spanning the 1st-17th centuries CE in Europe and North Africa. Analyses of our newly reported sequences, together with published mitochondrial DNA sequences, confirm that black rats were introduced into the Mediterranean and Europe from Southwest Asia. Genomic analyses of the ancient rats reveal a population turnover in temperate Europe between the 6th and 10th centuries CE, coincident with an archaeologically attested decline in the black rat population. The near disappearance and re-emergence of black rats in Europe may have been the result of the breakdown of the Roman Empire, the First Plague Pandemic, and/or post-Roman climatic cooling.

A full list of author affiliations appears at the end of the paper.

The black rat (*Rattus rattus*) is one of three rodent species, along with the house mouse (*Mus musculus*) and brown rat (*R. norvegicus*), to become globally distributed thanks to a close commensal relationship with humans[1]. Collectively, these taxa are highly significant to human societies both as pests responsible for billions of euros of damage to food stores annually[2], and as vectors and/or reservoirs that have contributed to the spread of numerous diseases, most famously bubonic plague[3,4].

Despite the significance of this rodent, our knowledge of the black rat's evolutionary history and taxonomy remains limited. Previous genetic studies have described a *R. rattus* complex involving multiple recognised species with potential introgression among different lineages[5–7]. Mitochondrial DNA studies have helped to resolve the taxonomic controversies by linking a monophyletic mitochondrial lineage to specific South Asian (now globally distributed) black rat populations that possess a $2n = 38$ karyotype (previously referred to as lineage I)[8–10]. The Asian house rat (*R. tanezumi*), endemic to Southeast Asia, has been identified as the closest sister group of the black rat (previously designated as lineages II through IV). The divergence between the two species has been dated to ~0.4 Mya[11], and the two have been suggested to hybridise[6,7,12].

The ability of rats to colonise, and become dependent upon anthropogenic niches[13] makes them ideal bioproxies to track historical processes[1,14,15]. Archaeological specimens of rats and mice have thus been used to track human migrations, trade, and settlement types in a wide range of contexts[16–22]. Previous archaeological and genetic evidence suggests that the pre-commensal distribution of the Eurasian black rat (based on the taxonomic definition proposed by mitochondrial DNA studies[8,9] and hereafter referred to as black rat, see SI for discussion) was largely limited to South Asia[10,23,24]. Black rat finds from cave sediments in the Levant spanning the late Pleistocene to early Holocene indicate a possible western distribution[25,26]. These remains, however, require direct dating to confirm their age, and there is a subsequent absence of rats from settlement sites in this region until at least the second millennium BCE[26].

The earliest large concentrations of presumed commensal rat remains reported thus far derive from late third, or early second millennium BCE settlements in both the Indus Valley and Mesopotamia[26]. Commensal black rats may also have reached the Levant and eastern Mediterranean region by the start of the first millennium BCE[26]. Based on archaeological evidence from Corsica, the Balearics, Italy and Morocco[27–29], black rats likely first appeared in the western Mediterranean basin towards the end of that same millennium.

The black rat's colonisation of Europe has been linked to the historical development of urbanism and trade networks, and their arrival is important for understanding historical plague pandemics including the 6th century Justinianic Plague and the 14th century Black Death[4,30–32]. The central role traditionally attributed to black rats and their fleas in the spread of the plague bacterium (*Yersinia pestis*) during these pandemics has been challenged on various grounds, however, including the historical distribution and abundance of rats, and this correlation continues to be debated[33–37].

Although surveys of zooarchaeological rat finds from archaeological sites across Europe suffer from considerable regional variation in coverage, the available data indicates successive episodes of dispersal north of the Mediterranean associated first with Roman expansion (first century BCE to second century CE), and then with the emergence of medieval economies from the 9th century CE, punctuated by a decline and a possible range contraction[32]. Black rat remains are found throughout the Roman Empire in the 1st–5th centuries CE, but rarely beyond its northern borders, suggesting that these rats were dependent on a Roman economic system characterised by a network of dense settlements connected by bulk transport via efficient road, river, and maritime routes[4,31].

With the breakdown of the Roman Empire from the 5th century onwards, evidence for the existence of black rats becomes scarcer. They may have been extirpated entirely from the northern provinces including Britain[32,38,39], and the percentage of archaeological sites with black rat remains declined even in the Western Empire's Italian core[40]. By contrast, black rats remained common in the Balkans and Anatolia until at least the 6th century CE, presumably reflecting continued stability in the Eastern Roman Empire[41–44]. Since zooarchaeological data between the 5th–8th centuries is limited in many European and Mediterranean regions, the pattern of post-Roman absence may partly represent research bias[45], though early medieval black rats are rare even where other small mammals are reported[38].

Black rats reappeared at northern European trading settlements during the 9th and 10th centuries CE, including sites well beyond their prior Roman range, including Hedeby in northern Germany and Birka in Sweden, as well as former Roman towns and high-status early medieval settlements such as York and Flixborough in England and Sulzbach in Bavaria[46–50]. The subsequent expansion of urbanism and large-scale trade of bulk goods in medieval Europe appears to have favoured rats, just as in the Roman period. By the 13th century CE, black rats were present throughout most of Europe[4] and they reached southern Finland by the late 14th century[51]. Black rats remained widespread across Europe until at least the 18th century, before their population declined, most likely as a result of competition with the newly arrived brown rat, the rat species that is now dominant in temperate Europe[52–54].

It remains unclear whether the black rat was actually extirpated from post-Roman northern and western Europe; and whether medieval rat populations in temperate Europe derived from the remnant population in southern Europe, or from another wave of rats that were introduced from beyond the Mediterranean (e.g. via Rus' river trade[32,55]). These questions are relevant to several key debates in European economic and environmental history including: (1) the extent to which the end of the Western Roman Empire represented a crisis in urbanism and trade—particularly in bulk goods such as grain—as well as a political collapse[56–58]; (2) the role of easterly vs. westerly connections in the rise of northern European medieval urban networks[59,60]; and (3) the model of the spread of the Justinianic Plague and the subsequent First Pandemic. This pandemic started in the eastern Mediterranean in 541 CE, spread quickly across Europe including England, and continued for approximately two centuries[61–63], a period that coincides with the gap in archaeological evidence for rats in northwest Europe. Given the limitations of both zooarchaeology and genetic studies of modern rat populations to address successive waves of contact after a species is established, ancient DNA may help to resolve these questions by directly revealing the presence or lack of genetic continuity through time.

Here, we employ a nested three-stage approach to address these questions. First, we assemble a de novo reference genome of the black rat. This genome allows us to investigate the long-term demographic history of the black rat in relation to other rat species, and to establish the foundation for genome-wide analyses of ancient remains. Second, we report 70 new mitochondrial genomes from 3 modern and 67 ancient European and North African black rats, including archaeological specimens spanning the Roman to early post-medieval periods (1st–17th century CE). We explore the dispersal of black rats into the Mediterranean and Europe by analysing these alongside 132 mitochondrial DNA sequences (106 historic and 26 modern) generated from modern

| Table 1 Assembly statistics of the de novo black rat reference genome. | |
|---|---|
| Scaffold number | 6805 |
| Scaffold N50 (Mb) | 145.8 |
| Largest scaffold (Mb) | 260.8 |
| Assembly size (Gb) | 2.25 |
| Scaffold length >10 Mb (Gb) | 2.23 |
| GC content (%) | 42.1 |
| Repetitive region (%) | 38.4 |

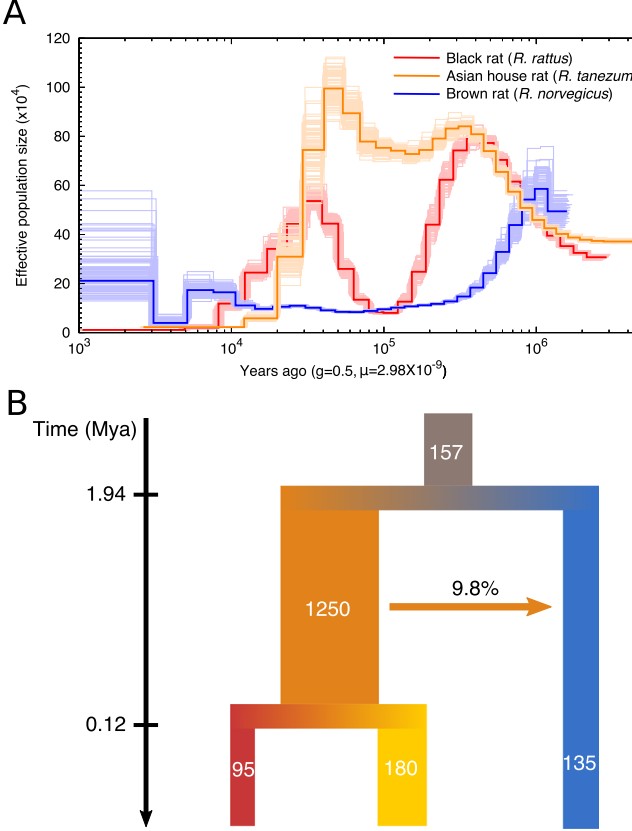

**Fig. 1 The demographic history of the black rat and its closely related species. A** Population dynamics of the black rat (*R. rattus*), Asian house rat (*R. tanezumi*) and brown rat (*R. norvegicus*) estimated by PSMC, with 100 bootstrap replicates. **B** Demographic modelling of the divergence and migration among the black rat, Asian house rat and brown rat estimated by G-PhoCS. The values represent the average estimates of effective population sizes (in thousands), population divergence times (Mya) and the total migration rate through time. The 95% HPD range of all estimates are listed in Supplementary Data 1.

and museum black rat specimens from across western Eurasia, the Indian Ocean, and Africa. Lastly, we generate 36 nuclear genomes from our archaeological black rats and three from modern black rats and use these to explore the species' population history in Europe and the Mediterranean from the 1st to 17th centuries CE, focusing particularly on the hypothesis of dual dispersals in the Roman and medieval periods. We then interpret the black rat's dispersal history within the context of major historical processes.

## Results

**The demographic history of the black rat and its closely related species.** To facilitate the study of the demographic history of the black rat, both before and after the establishment of its commensal relationship with humans, we first generated a de novo genome assembly of the black rat using a wild-caught individual from California, USA (Supplementary Note 1). Combining shotgun, Chicago and Hi-C sequencing data with the Dovetail HiRise assembler pipeline[64], we obtained a genome assembly with a total length of 2.25 Gb and a scaffold N50 reaching 145.8 Mb (Supplementary Tables 1 and 2). The 22 scaffolds with over 10 Mb covering 98.9% of the entire assembly (Table 1), with each of the 18 autosomes corresponding to one large scaffold each and over 90% of the X chromosome represented by four scaffolds (Supplementary Figs. 1 and 2, Supplementary Note 1). The average GC content is 42.1%, similar to the brown rat reference genome *Rnor_6.0* (42.3%), and 38.4% of the assembly was identified as repetitive elements (Supplementary Table 3). Benchmarking Universal Single-Copy Orthologs (BUSCO) analysis[65] also revealed a high completeness of this genome assembly, with 90.1% complete BUSCOs identified using eukaryotic dataset, comparable with *Rnor_6.0* (91.4%) (Supplementary Table 4). Because the Asian house rat and the black rat are both present in California, we also assessed potential introgression from the Asian house rat into our black rat individual. A signature of introgression would limit the value of our de novo genome as a reference genome onto which reads derived from ancient black rats could be mapped. Our analyses suggested no significant introgression signature in the Californian black rat (Supplementary Note 1, Supplementary Table 5).

To address the demographic history of black rat, we applied the pairwise sequentially Markovian coalescent (PSMC)[66] analysis to estimate its population size dynamics alongside the brown rat and Asian house rat. When calibrated with a mutation rate of $2.96*10^{-9}$ per generation and generation time of 0.5 years[67], the analyses revealed different dynamic patterns of population size changes amongst these rat species (Fig. 1A). The brown rat experienced a population decline beginning ~1 Mya, as described previously[67], while both the black rat and Asian house rat populations expanded until 300–400 thousand years ago (kya). The black rat population then experienced a bottleneck with an 8-fold drop in effective population size until 100 kya, and a re-expansion from 100 to 40 kya. The Asian house rat, however, did not experience a population decline until ~40 kya, when both black rat and Asian house rat populations experienced declines that have continued to the present.

To investigate the population sizes, split times, and migrations among these rat lineages, we applied Generalised Phylogenetic Coalescent Sampler (G-PhoCS)[68]. The result revealed a similar population size dynamism, with the effective population size (Ne) of black rat/Asian house rat ancestral lineage estimated to be $1.25*10^6$, about tenfold the Ne of black rat/Asian house rat/ brown rat lineages (Fig. 1B, Supplementary Note 2, Supplementary Data 1). The split time between brown rat and black rat/ Asian house rat lineages was estimated to be 1.94 Mya (within the 95% highest posterior density (HPD) range estimated using mitochondrial genomes in a previous study[11]), while the split of Asian house rat and black rat lineages took place ~120 kya. This recent split time relative to the coalescent time estimate based on mitochondrial genomes between these two lineages could be explained by the large ancestral population size of the black rat/ Asian house rat lineage[68]. Among these lineages, we only detected one instance of gene flow from the black rat/Asian house rat ancestral lineage into brown rat lineage, with an introgression proportion of 9.8%.

Taken together, we observed population expansions and bottlenecks in the black rat during the last million years, and a smaller Ne relative to the Asian house rat. This could be explained by the relatively limited geographic distribution of the black rat in southern Asia before the initiation of its commensal relationship with people, and the fact that the Asian house rat is endemic to a much greater area in southeastern Asia[10]. We did not detect any genomic introgression between the lineages leading to the black and Asian house rat, suggesting these two species were geographically isolated after their split from a common ancestor for a sufficiently long period to facilitate their reproductive incompatibility.

**A global phylogeography of the black rat based on mitochondrial DNA.** We collected 191 ancient black rat individuals from 33 archaeological sites across Europe, North and East Africa, and southern Asia dating from the 2nd millennium BCE to the 17th century CE, plus eight modern individuals from North Africa (Supplementary Note 3, Supplementary Data 2). After shotgun screening, we retrieved 70 mitochondrial genomes (67 ancient and 3 modern, with coverage spanning 3.5×–300.0×) from samples from 18 sites in Europe and the Mediterranean (Supplementary Data 2), and identified 40 haplotypes. The phylogenetic tree based on mitochondrial genomes revealed two clades: a major clade with 32 haplotypes from 47 ancient and modern samples, together with the modern Californian rat, and a minor clade consisting of eight haplotypes and 23 ancient samples from the 6th-century site of Caričin Grad, Serbia (Supplementary Fig. 3). The phylogenetic resolution within each major clade was relatively poor, though samples from the same or closely related

sites occasionally formed sub-clades including the samples from modern-day Zembra (Tunisia) and medieval central Europe.

In order to establish the relationship between the ancient rats and modern black rats from across their range, we analysed the cytochrome b (CYTB) region from 476 samples, including our 67 ancient samples and three modern samples alongside 132 previously unpublished modern and historic CYTB sequences from across the Indian Ocean basin (Supplementary Data 2), and 274 sequences published in previous studies[10,24,69,70]. The maximum-likelihood tree of the CYTB region revealed that all the ancient rats from this study belong to the previously described black rat lineage I (Fig. 2, Supplementary Figs. 4, 5 and Supplementary Data 2). None of the ancient rats from this study fell into *Rattus* lineages II–VI. Within lineage I, we recapitulated the unnamed substructure and assigned the terms A–E to the five major haplogroups[10]. In addition to these five, we confirmed a sixth lineage I haplogroup, F, consisting of modern samples from Sri Lanka and the Andaman Islands, which is basal relative to all other lineage I black rats and has previously been reported as the Sri Lankan unique sub-lineage, RrC LIb[71] (Fig. 2, Supplementary Fig. 4B, Supplementary Data 2).

Haplogroup A within lineage I (previously described as the European ship rat[24]) was the most common among the analysed samples (179/354). Members of this haplogroup include ancient and modern black rats from Europe and regions of the world with a history of colonisation by, and/or trade with European powers. The only additional haplogroup found in Europe was Haplogroup C (previously described as the Arab ship rat[24]) at Caričin Grad, Serbia, which included 24 archaeological individuals. Haplogroup C is found in modern rats from India, Egypt, East, South and

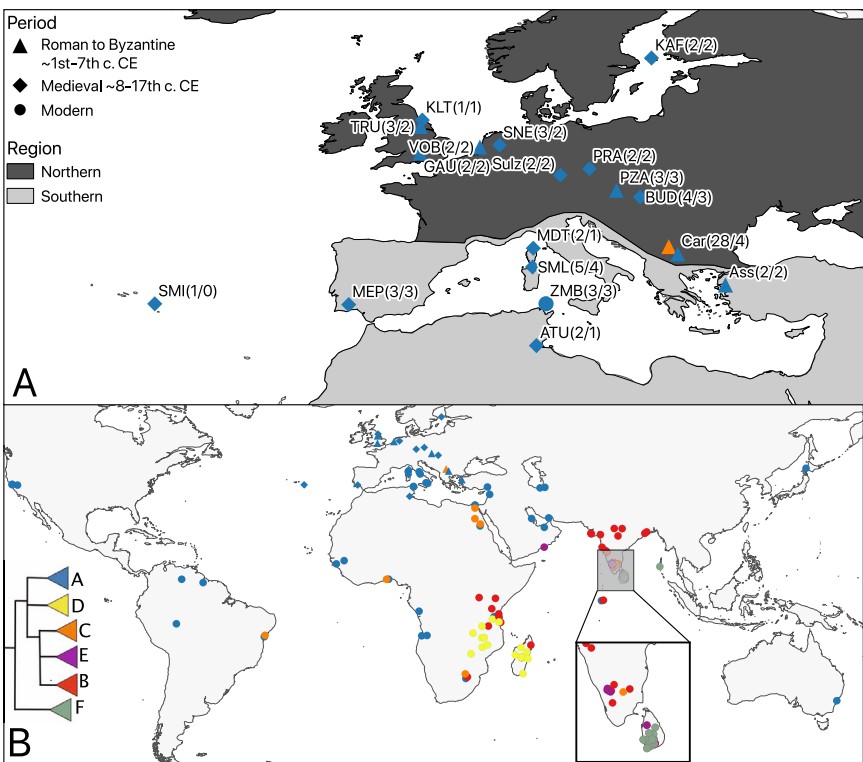

**Fig. 2 Sampling sites and mitochondrial phylogeographic patterns. A** Map of sampling locations. The numbers in parentheses are numbers of samples included in mitochondrial/nuclear genome analysis. SMI (Villa Franca do Campo), MEP (Mertola), KLT (Kilton Castle), TRU (Tanner Row, York), GAU (Gatehampton Villa), VOB (Voorburg-Forum Hadriani), SNE (Deventer-Stadhuiskwartier), MDT (Monte di Tuda), SML (Santa Maria Lavezzi), ATU (Althiburos), Sulz (Castle Sulzbach), PRA (Prague Castle), PZA (Petronell-Carnuntum Zivilstadt), KAF (Kastelholm), BUD (Buda Castle-Teleki Palace), Car (Caričin Grad) and Ass (Assos). **B** The phylogeographic pattern of black rat revealed by CYTB mitochondrial haplogroups (see Supplementary Fig. 4 for detailed phylogeny), including 67 ancient rats, 3 modern rats from Zembra, 132 modern samples from Indian Ocean basin and 274 published samples. Basemap source: ESRI ArcWorld Supplement (World Continents), used under license.

West Africa, and South America. None of the other haplogroups were present in Europe or the Mediterranean region. Haplogroups B and E only included modern samples from India and countries bordering the Indian Ocean. Haplogroup D (previously described as the Madagascar and Indian Ocean islands group[24]) included primarily samples from Madagascar and East Africa, and Haplogroup F consisted of samples from Sri Lanka and the Andaman Islands (Fig. 2, Supplementary Data 2).

To investigate the introduction route of black rats into Europe, we analysed mitochondrial cytochrome B sequences derived from globally distributed modern and ancient black rats. Previous studies indicated that the black rat originated in the Indian Peninsula[10,24,69,72]. Leaving aside the putative Late Pleistocene to early Holocene records from the eastern Levant, the earliest finds of presumed commensal black rats derive from the Indus Valley and Mesopotamia in the 3rd/2nd millennium BCE, coincident with the emergence of urbanism and establishment of trade links between these regions[26,73], though a more westerly limit to the black rat's natal range cannot be excluded. The source for dispersal to the Mediterranean and ultimately Europe remains unclear. Suggestions include maritime trade from India and/or the Arabian peninsula into the Red Sea and subsequently through Egypt (perhaps via the canal built under Darius[74]) in the mid/late first millennium BCE[4,32], or more likely earlier overland communication routes between Mesopotamia and the Levant[26,73].

While a maritime route is clearly implicated in the black rat's dispersal to East Africa[75,76], our results tentatively favour an overland hypothesis for its dispersal from South Asia to the Mediterranean to Europe, since both ancient and modern black rats from Europe and the eastern Mediterranean share haplogroups with sampled populations from Iran and the Persian Gulf, but not with Indian Ocean samples from southern India to Madagascar (Fig. 2). The results also suggest a secondary dispersal route via Egypt, given the appearance of Haplogroup C at the 6th century CE Byzantine site of Caričin Grad, Serbia and in modern samples from the Nile valley. While hypothetical, this might reflect Egypt's central roles both in direct Indo-Roman trade, following its annexation in 30 BCE, and in grain production for the Roman and early Byzantine Empires[4,77]. To test these hypotheses, further investigations into ancient and modern black rat populations from the Levant, Mesopotamia, Egypt and the Indus Valley are necessary.

**Ancient genomes reveal the relationships of European black rats over space and time.** To explore the black rat's European population history in greater detail, we shotgun sequenced 36 ancient and three modern black rats from 17 sites to 0.2×–16× coverage for whole genome analysis, including 18 females and 21 males determined by the coverage on sex chromosomes (Supplementary Data 2). The deeper sequenced ancient samples spanned two broad time periods, including 15 from the Roman and Early Byzantine period (1st–7th century CE), and 21 from medieval and post-medieval contexts (8th–17th century CE) (Supplementary Data 3). Geographically, all the samples were divided into two groups: a "northern" group of 25 samples from temperate Europe, and a "southern" group of 11 samples from the Mediterranean and Portugal (Fig. 2). After mapping and genotyping, we identified 7,869,069 bi-allelic transversion variants in the autosomal non-repetitive regions. They were combined with the Californian black rat for de novo genome assembly, and two published brown rats and one published Asian house rat for downstream population genomic analyses.

The phylogenetic tree constructed from autosomal SNPs revealed complex relationships among ancient black rats from different regions and time periods (Fig. 3A). Except for the late

medieval (ca. 14th century) to Ottoman (ca. 17th century) site of Buda Castle, Hungary, samples from the same site are clustered together. All the samples from the northern group, together with one southern sample from the medieval period—from 8th to 9th century Althiburos, Tunisia—formed a single clade, while all the other Byzantine to medieval samples from the southern group formed several separate clades consistent with their local geographic region. The rats from the southern group also possessed higher heterozygosity than those from the northern group, within both Roman/Byzantine and medieval/post-medieval periods (Supplementary Fig. 6, Supplementary Table 7, Supplementary Data 4). This could be explained by the longer history of rats in the Mediterranean which date to at least the first millennium BCE[26,27], and the founder effects of limited introductory waves of rats into the northern region.

Within the major northern cluster, samples were divided into two smaller clusters representing Roman/Byzantine and medieval/post-medieval periods, respectively. The only exception was a medieval Tunisian sample that falls into the Roman cluster. Within each cluster, samples grouped together based upon their geographic location (central Europe, western/northern Europe, Serbia). These phylogenetic relationships suggest that the initial black rat population in temperate Europe was replaced by a genetically distinct population after the 6th century CE. The younger population was first documented in early medieval (8th to early 10th century CE) Sulzbach, Germany. The Roman-like gene pool was still present during the 8th–9th century in North Africa, though due to the lack of more recent samples we cannot address whether or when the second wave arrived there. A similar pattern was also revealed by multidimensional scaling (MDS) based on isolation-by-state (IBS) distance among the samples (Supplementary Fig. 7).

The phylogenetic tree based on Y-chromosome scpMSY regions (Supplementary Data 5) similarly demonstrated that the Roman rats formed a single cluster. However, unlike the autosomal phylogeny, all the post-Roman samples from both the northern and southern groups, including Byzantine Assos and Caričin grad, formed a separate cluster (Supplementary Fig. 8), without well-supported substructures. Given the male-biased dispersal pattern commonly described in the black rat and other rodent species[78,79], this might indicate a male-specific replacement that took place in both the Mediterranean regions and temperate Europe.

A decline in the European black rat population during the 6th–9th centuries has previously been suggested based on zooarchaeological evidence[32,38,39]. This has been attributed to several causes including: (a) the demise of the Western Roman Empire's economic and urban system from the 5th century CE, including the cessation of large-scale grain shipments that may have helped to disperse and support rat populations[4]; (b) climatic cooling in the 'Late Antique Little Ice Age'[80]; and/or (c) the Justinianic Plague, which began in 541 CE and is likely to have infected rat populations previously naive to *Yersinia pestis*, regardless of their potential role in its spread among humans[4,81,82]. Our finding of a post-6th-century turnover corroborates this apparent decline, though the density of our samples' spatiotemporal coverage is not sufficient for us to distinguish between the potential causes. To understand how the Justinianic Plague influenced the rat population, further studies should focus on archaeological black rats from contexts postdating the mid-6th century in areas of the Byzantine Empire and wider Mediterranean where an urban settlement system persisted.

The medieval Tunisian (Althiburos) sample indicates a different population history of black rats in North Africa relative to temperate Europe. Black rats from a wider range of time periods resident in North Africa and the western Mediterranean would allow us to test whether there was continuity within the

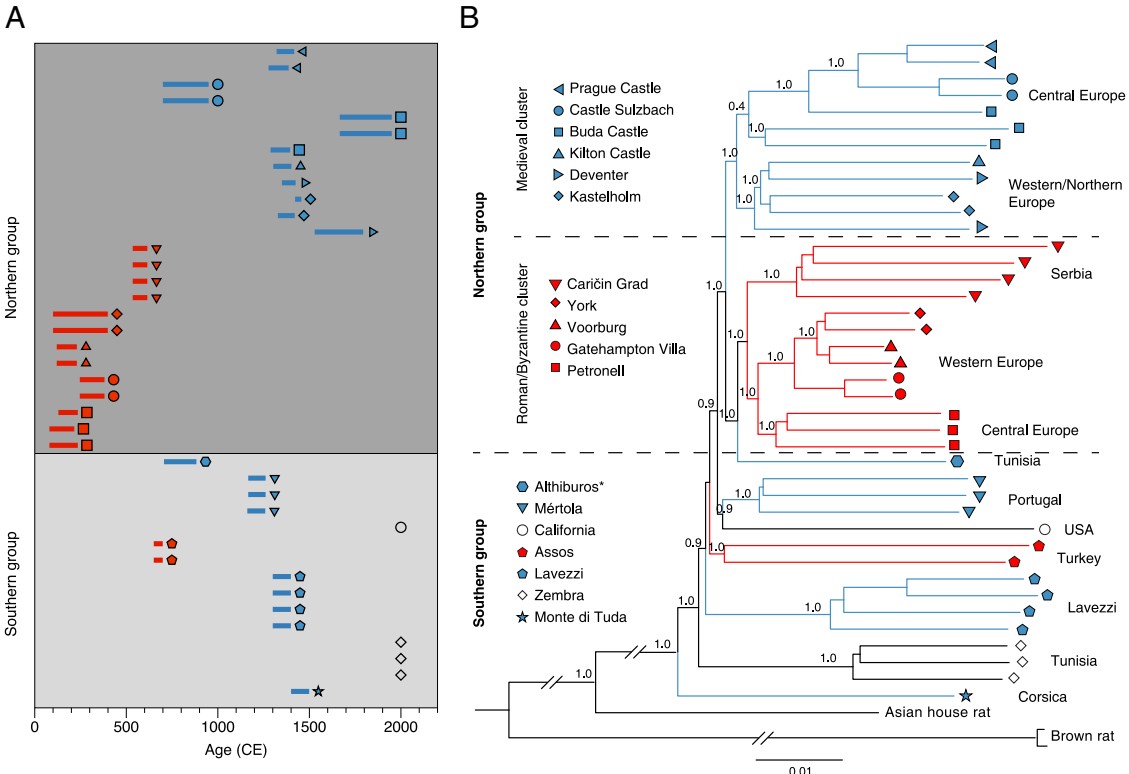

**Fig. 3 Relationships of the ancient black rats over time and space. A** The ages of the rat samples included in whole genome analyses. The bars represent 95.4% confidence intervals surrounding the direct radiocarbon dates or stratigraphic dates (Supplementary Data 3). The colours correspond to the Roman/Byzantine (red) and medieval (blue) time periods. The symbols represent the sampling sites listed in panel B, and the modern samples are represented by black symbols. **B** The phylogenetic relationship among ancient and modern black rats reconstructed using a neighbour joining phylogeny. The pairwise genetic distances were calculated using autosomal variants. The support values based upon 100 bootstrap replicates are shown on the nodes. The branches are coloured by sample ages as described in panel **A**, and the tip symbols correspond to the sampling site. * Though the medieval Tunisian (Althiburos) sample clusters geographically in the southern group, it falls in with the Roman cluster of the northern group in the phylogeny.

black rat populations from the Roman to early Islamic period (ca. 8th century). This is particularly pertinent to debates concerning the degree of continuity between the Roman Empire and the Early Islamic world, notably in urban settlements and trade networks[83].

To investigate the genetic interaction between different rat populations further, we applied a series of $f$-statistics. Based on the result of the $f_4$-statistics symmetry test, the ancient samples were divided into 18 groups (Supplementary Data 4). Of these, 16 correspond to samples from 16 different sites, while the three late/post-medieval samples from Buda Castle (Hungary) fell into two groups corresponding, respectively, to late medieval (14th–15th century) and Ottoman (17th century) periods (Supplementary Data 3).

First, we investigated if any Roman population contributed to the Byzantine or medieval groups, with $f_4$(norvegicus, Byzantine/medieval; Roman1, Roman2). We found that of two Roman geographical groups (central European represented by Austria, and western Europe represented by Britain and the Netherlands) the western rats were significantly more closely related to all the Byzantine and medieval groups (Fig. 4A, Supplementary Fig. 9, Supplementary Data 4). This result suggests that despite the population turnover that occurred in temperate Europe after the Roman period, Roman black rats from western Europe may have contributed to populations that colonised temperate Europe following the decline of the original population.

Next, we applied $f_4$(norvegicus, Roman; Byzantine/medieval1, Byzantine/medieval2) to test if there were any differences in the relative contribution of Roman rat ancestry into the Byzantine or

medieval populations. In agreement with the phylogenetic and MDS analysis, most northern groups were significantly more closely related to the Roman rat populations compared to the Byzantine or more recent southern groups (SML, MDT, Ass). The lone exception to this pattern were two post-medieval samples from Buda Castle (BUD001/4), which were equally related to the Roman groups and the Assos (Ass) group that consists of two samples from Byzantine Turkey (Supplementary Data 4). Among the northern groups, the medieval rats from Åland (Finland), the UK, and the Netherlands, as well as Byzantine rats from Serbia, were more closely related to the Roman rat populations than were medieval rats from central Europe (represented by populations in Germany, Czech Republic, and Hungary). This suggests that the genetic contribution from putatively western European Roman rats, was also greater in the local western European medieval rats.

We also investigated the relationship between the Buda Castle (Hungary) samples from different time periods by contrasting them with the other medieval rats from temperate Europe (Fig. 4A, Supplementary Fig. 9, Supplementary Data 4). As revealed by the phylogenetic tree, both the German and Czech rats shared more genetic affinity with the ca. 14th–15th century Buda Castle (BUD003) sample, than with the 17th century or later specimens (BUD001/4). Having said that, BUD001/4 still showed higher affinity to BUD003, when compared with all other populations. This evidence suggests a black rat population transition in this region between the 14th/15th century and the late 17th century, potentially related to the 16th–17th century Ottoman occupation of Buda (Hungary), while the local medieval ancestry was still present in the later population.

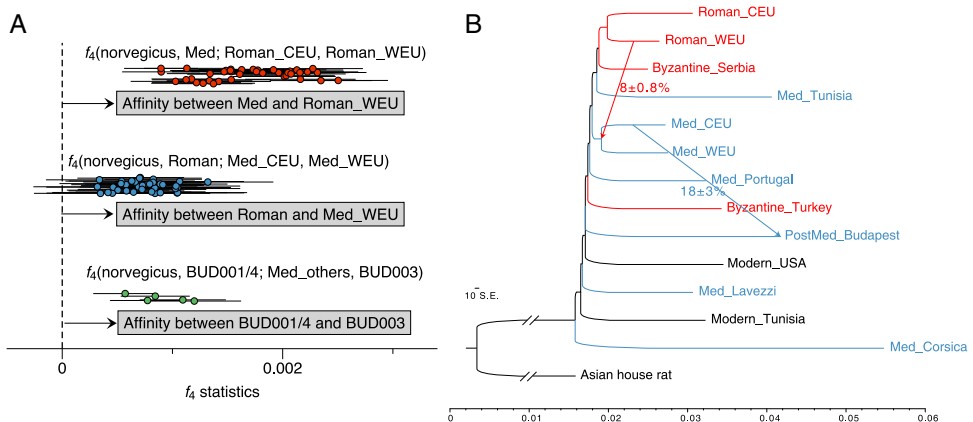

**Fig. 4 Gene flow among ancient rat populations. A** The *f*-statistics showing admixture between different ancient rat populations. The dots show all the combinations of $f_4$-statistics as described above each cluster, with the center being $f_4$ values and the error bars being ±3SE of the estimates. The SE is determined using jackknife with sample size (*n*) being the number of autosomal scaffolds. The three clusters show the affinity between: (top, red) medieval rats (Med) and western European Roman rats (Roman_WEU); (middle, blue) Roman rats (Roman) and western European medieval rats (Med_WEU); and (bottom, green) post-medieval Buda Castle rats (BUD001/4) and the medieval Buda Castle rat (BUD003), respectively. The source data is provided in Supplementary Data 4.2, 4.3 and 4.4. **B** Admixture graph with two migration events fitted, estimated by Treemix. The migration edges are displayed by arrow including the introgression fractions and standard errors. The colour of each branch represents the time period of each group: Roman/Byzantine (red) and medieval/post-medieval (blue).

To corroborate the patterns of gene flow suggested by the $f_4$-statistics, we used Treemix[84] to generate an admixture graph of all ancient rat populations, using the Asian house rat as an outgroup. The maximum-likelihood population tree without any admixture produced a similar topology to the neighbour-joining autosomal phylogeny (Supplementary Fig. 10). The rats from the northern group and a medieval Tunisian rat formed a clade, to which all the other Mediterranean rats were an outgroup, without any significant clustering pattern among the lineages. When admixture events were allowed, the first two suggested gene flow edges were from the medieval central European population into the post-medieval Buda Castle population, estimated to 18.2 ± 3.0%, and from the Roman western European population into the ancestral lineage of the medieval European populations in the northern group, estimated to be 8.1 ± 0.8% (Fig. 4B).

The results from both the *f*-statistics and Treemix analyses revealed a degree of Roman rat ancestry in the medieval populations. More specifically, medieval rats were more closely related to the Roman rat populations from the Netherlands and Britain (Fig. 4, Supplementary Data 4). This signal suggests a reservoir of black rat population in western Europe that admixed with the re-introduced medieval population. The stronger affinity of medieval western European populations to Roman populations (Supplementary Data 4) also suggested that this relict population was more likely distributed in western and not central Europe. This result could indicate that rats from the northernmost Roman provinces were not extirpated, despite their absence in early medieval zooarchaeological assemblages. Alternatively, and in our view more likely, the inferred relict population may have been located in an unsampled region of France or the Iberian Peninsula. The observation that medieval rats from temperate Europe fall into the same cluster as Roman rats also suggests that the second (medieval) wave of introduction to temperate Europe probably originated from the same source population as the first (Roman) dispersal. Considering the zooarchaeological evidence that rat populations in southern Europe persisted after the collapse of the Western Roman Empire, notably in Italy[40], it is likely that southern Europe was the source of reintroduced rats in temperate Europe.

Given the presence of rats in 9th century northern *emporia* (proto-urban trading sites) around the North and Baltic

Seas[46,49,50], a southern European origin would emphasise the importance of the Carolingian Empire (the Frankish polity which controlled much of western and central Europe as well as northern Italy in the 9th century CE) and routes such as the Rhône and Rhine corridors in reestablishing large-scale trade links between the Mediterranean and northern Europe[85]. This connection remains tentative until samples from the early emporia themselves, mainland Italy, and the Iberian Peninsula can be investigated. Samples from the early Islamic world derived from the Iberian Peninsula and North Africa would also help to clarify the population history of black rats.

**Discussion**

This study explores the historic dispersal of commensal black rats using a de novo genome assembly for the black rat, ancient and modern mtDNA from across Europe, Africa, and the Indian Ocean, and ancient nuclear genomes from the Mediterranean and Europe. Our results confirm that the black rat was most likely introduced to the eastern Mediterranean by an overland route through Southwest Asia, though a maritime route via the Indian Ocean and Red Sea cannot be excluded. We identify two waves of rat introduction into temperate Europe. The first likely accompanied the Roman northward expansion during the first centuries BCE/CE and the second took place during the medieval period (starting in the 8th–10th centuries CE). The rats in this second wave were probably derived from the same ancestral population as the first, and subsequently admixed with a western or southern European relict population from the first wave.

Considered alongside the paucity of archaeological rat remains from the 6th to 8th centuries CE (particularly in northern and western Europe), this population turnover suggests that the black rat population and range declined during the early medieval period. This may have been associated with the breakdown of the Roman Empire— from the 5th century CE in western Europe and the early 7th century CE in the Balkans—and with it the network of well-connected settlements that had previously supported black rat populations. Grain shipments may have played a key role in the dispersal and maintenance of rat populations during the Roman period, and it is notable that weevils (*Sitophilus granarius*) and other grain pests show a similar pattern of Roman introduction, apparent

post-Roman extirpation, and a medieval reintroduction in the former northern provinces[86]. Alternatively, or additionally, European rat populations may have been negatively impacted by the First Plague Pandemic and/or the climatic cooling of the Late Antique Little Ice Age, both of which began in the mid-6th century CE. To disentangle these scenarios, further zooarchaeological and genomic studies of ancient rats are required that span these centuries across a wider geographic range.

The medieval introduction of rats into Northern Europe is attested by their presence in Germany in the early 10th century (at the latest), coincident with an increase in the appearance of rat bones in archaeological contexts across the continent. Our results suggest a repopulation of temperate Europe from the south, perhaps linked with the development of trade routes in Carolingian western Europe, and probably not via early Russian riverine trade as has been previously hypothesised[32]. Black rats appear to have been a continuous presence in Europe from this point until the post-medieval period, spanning the 14th century Black Death and extending into the 17th century. This population may also have been supplemented by localised introductions, including one potentially associated with the Ottoman occupation of Buda from 1541 CE.

The recent dramatic decline in black rat populations across modern Europe, with local extirpation in much of the north of the continent, is likely linked to competition with the brown rat which arrived from Asia in the early 18th century[52,53]. The genetic and demographic impact of this dispersal on black rats is an important area for future investigations since by the late 18th century, naturalists in many European countries had already attributed a marked decline in the black rat to competition from the brown rat[87–90]. The black rat's significantly reduced, but persistent presence, particularly in towns, suggests a degree of niche partitioning between the two species[91].

Our results reveal the degree to which human-commensal species can undergo population dispersal and demographic fluctuations. In fact, because these dynamic evolutionary processes are tightly correlated with the characteristics of the human niche, commensal species can act as ideal proxies to interpret the history of human movement and cultural change.

## Methods

**Ethics statement**. Ethics board approval was not required for analysis of the black rat specimens since these were non-human and in most cases deceased upon collection. The exception, a modern black rat from California, was sourced from municipal pest control and humanely euthanized using a standard American Veterinary Medicine Association-approved method. In most cases, permits were not required for analysis of the ancient specimens, which were provided by co-authors on the paper who either excavated or already had access to the material. Specimens from Assos (Ass_1, Ass_2) were exported with the permission of the Çanakkale Museum, Turkey (77366169-152/766). Specimens from Rirha (AJ410-AJ414) were exported under a permit from the Ministry of Culture of Morocco. Rat bones from Panga Ya Saidi (AJ37-AJ38) and Chombo (AJ40-AJ41) were obtained and exported with permission from the National Council for Science and Technology, Kenya (Research Clearance Permit NCST/RRI/12/1/SS/541; Exploration/ Excavation Licence NMK/GVT/2) and National Museums Kenya (export permit dated 12/09/2011). The sample from Songo Mnara (AJ40) was exported under permit EA.402/605/01/9 issued by the Antiquities Division, Ministry of Natural Resources and Tourism, Tanzania. Formal loan/sampling agreements were signed with: University of Tartu Archaeological Collection (samples AJ363, AJ366; Sampling Protocol #89), South Holland Provincial Archaeological Depot (samples AJ469-AJ472; loan number 2018-27), Åland Museum (samples AJ404-AJ409, ÅLR 2018/3788). Recent, non-archaeological specimens were sampled from existing museum collections.

**Radiocarbon dating and calibration**. Fourteen ancient rat bones were directly radiocarbon dated via accelerator mass spectrometry (AMS) on bone collagen at Manheim (MAMS), University of Waikato (Wk), and Oxford University (OxA). These were analysed alongside two previously published dates from Gatehampton[92] (Supplementary Data 3). One additional sample (SNE002/Wk-51521) failed due to insufficient collagen. All radiocarbon dates were calibrated in OxCal 4.4[93], using the IntCal20 calibration curve[94].

Given the omnivorous diet of black rats, stable isotope values were monitored for evidence of marine dietary contributions that might result in significant reservoir effects. Where possible, $\delta^{13}C$ (‰, VPDB) and $\delta^{15}N$ values (‰, AIR) were obtained by the respective dating laboratories using Isotope Ratio Mass Spectrometry (IRMS) and their standard protocols; otherwise $\delta^{13}C$ values were used as reported from the AMS. Nitrogen isotope values were available for 10 specimens and fell between 6.9‰ − 11.9‰, consistent with published data for commensal brown rats[95] and with an omnivorous diet. Carbon isotope values ranged from −21.9‰ to −17.4‰. In the absence of detailed local comparative isotope data from terrestrial and marine species, it was not possible confidently to estimate marine dietary contribution and hence the magnitude of any required correction. Nonetheless, we performed indicative corrections for specimens whose $\delta^{13}C$ values suggested a possible non-negligible marine component in order to test for any possible impact on our interpretations. The cut-off for this was set as −18.5‰, based on published values for European terrestrial herbivores and the enrichment expected due to trophic level in an omnivore.

For six specimens with $\delta^{13}C > −18.5‰$, percentage marine contribution was estimated using "formula 1" from[96], with terrestrial and marine endpoints of −21‰ and −12‰, respectively, and a trophic enrichment factor of 1‰. These values were used to recalibrate the dates using mixed IntCal20 and Marine20[97] curves, and the magnitude of potential offset was assessed. In no case did the median calibrated date change by more than 140 years, and in no case would it have moved a specimen into a different chronological category or altered our interpretations. Given the uncertainty inherent in this process, the uncorrected ranges are used in Fig. 3, and details of the indicative corrections are given in Supplementary Data 3. In the majority of cases, the uncorrected date range coincided more closely with the stratigraphic dating.

**De novo genome assembly**. The black rat genome was sequenced and assembled using DNA extracted from the liver of a male wild-caught individual from California, USA. Shotgun, Chicago and Dovetail Hi-C libraries were prepared and sequenced on Illumina HiSeq 4000 platform and the genome was assembled using Meraculous 2.2.5.1[98] and HiRise scaffolding pipeline[64]. The detailed information of genome assembly is provided in the Supplementary Note 1.

The repetitive regions were identified using RepeatMasker 4.0.7[99] with Repbase 20170127 and the query species set as rattus, and TRF 4.09 (Tandem repeats finder)[100], with parameters set as "2 7 7 80 10 50 12". The completeness of genome assembly was assessed by BUSCO 3.0.2[65], using the 303 orthologs in the Eukaryota odb9 dataset. The new genome assembly was aligned to the brown rat reference genome *Rnor_6.0* using nucmer 4.0.0 in MUMmer tool package[101], to investigate the synteny between black rat and brown rat genomes, using both masked assemblies and anchor matches that are unique in both reference and query (Supplementary Note 1).

**Mitochondrial Cytochrome B fragment sequencing**. Overall, 292 tissue samples identified as the black rat were included for analysis, including 263 museum specimens (sampled from the collections at: American Museum of Natural History, Natural History Museum London, Field Museum Chicago) and 29 modern specimens collected in the field representing a wide geographic area at the periphery and on islands within the Indian Ocean.

DNA extraction and sequencing of these modern and museum samples were conducted in the modern laboratory at the Archaeology Department of Durham University, following standard protocols (Supplementary Note 4). The cytb region was amplified in 10 overlapping fragments and a variety of primer combinations was used depending on the nature of the sample (Supplementary Table 6). The sequencing reaction was carried out by the DNA Sequencing Service at the School of Biological and Biomedical Sciences at Durham University. The sequencing chromatograms were edited manually, subsequently assembled, and a consensus sequence per individual exported using Geneious R6 version 6.0.6 (https://www.geneious.com). Standard anti-contamination guidelines were followed. We successfully amplified cytb sequences from 202 of 292 samples. Only those that possessed >90% gene coverage were included in the analysis, which left 132 sequences.

**Ancient DNA extraction and processing**. We sampled 191 ancient black rats and eight modern black rat individuals from 33 archaeological sites across Europe and three sites in North Africa (Supplementary Note 3, Supplementary Data 2). Where multiple samples were taken from the same or related archaeological contexts, care was taken to ensure that these represented discrete individuals—either by sampling the same skeletal element and side, or on the basis of differing size and/or age.

Ancient DNA extraction was performed in dedicated ancient DNA facilities at the University of Oxford, the Max Planck Institute for the Science of Human History in Jena, and the University of York. All of the ancient lab facilities followed standard ancient DNA laboratory practices to minimise contamination, including the use of blanks at each stage from extraction to amplification. All material analysed at Oxford underwent the following treatment. Due to the small size of black rat bones, the outer surface of the bones was not removed prior to extraction. Bones that weighed <50 mg were completely consumed during the extraction process. The bone or tooth was cut using a Dremel drill with a clean cutting wheel

per sample (Dremel no. 409) and pulverised in a Micro-dismembrator (Sartorious-Stedim Biotech). Material analysed at York was subjected to bleach treatment (6% sodium hypochlorite for 5 min, and then rinsed with ultrapure water three times) prior to powdering following the same procedure as Oxford.

Extractions performed in Jena followed a silica-based protocol[102] using 50 mg of bone powder. Extractions performed at the University of Oxford were conducted using the Dabney protocol with a modification of the addition of a 30 min pre-digestion stage[103]. Extractions performed at the University of York were conducted using a silica-based protocol[104] modified to include a DNA concentration step using a centrifugal filter[105].

For each sample processed in Jena, a double-stranded DNA sequencing library was prepared from 20 μL of extract, with partial uracil-DNA-glycosylase (UDG) treatment (hereafter denoted as ds_halfUDG) or without UDG treatment (ds_nonUDG), following a published protocol[106]. Sample-specific index combinations were added to the sequencing libraries[107,108]. The indexed libraries were shotgun sequenced on an Illumina HiSeq 4000 instrument for screening, with 75 single-end-run cycles for ds_halfUDG libraries and 75 double-end-run cycles for ds_nonUDG libraries. After screening, one ds_nonUDG library and seven ds_halfUDG libraries were deep sequenced in the University of Kiel, on an Illumina HiSeq 4000 platform with 75 double-end-run cycles using the manufacturer's protocol.

All extracts generated at the University of Oxford and the University of York were built into Illumina libraries using double stranded methods using the Blunt-End Single-Tube Illumina library building (BEST) protocol as described previously[109] at the University of Oxford (ds_nonUDG). To amplify each library, 15 μL of library was added along with the following reagents: 25 μL of Accuprime Supermix I, 4 μL of BSA(10 mg/ml) and 3 μL of indexing primers 2 μM for a total volume of 50 μL. An additional barcode was added to the IS1_adapter.P5 adapter resulting in a double external indexed library. The libraries were then amplified on an Applied Biosystems StepOnePlus Real-Time PCR system, to determine both the success of the library build and the number of optimum cycles to use for the indexing PCR reactions. These 164 libraries were pooled at equimolar concentrations ready for sequencing. The pool of libraries was sequenced on an Illumina HiSeq 4000 (paired-end 75 bp) at the Danish National High-Throughput Sequencing Centre.

Ten extracts from Oxford were built into single-stranded libraries at the Max Planck Institute for Evolutionary Anthropology in Leipzig, Germany. The libraries were built from 30 μl of DNA extract in the absence of uracil DNA glycosylase (ss_nonUDG) followed by double indexing, using an automated version of the protocols described in[107,108] on a liquid handling system (Agilent Technologies Bravo NGS Workstation). From the initial screening run results 31 ds_nonUDG libraries from Oxford were included for deeper sequencing in Jena, together with the ten ss_nonUDG libraries, on an Illumina HiSeq 4000 platform at the Max Planck Institute for the Science of Human History, Jena with 75 single-end-run cycles (Supplementary Data 2).

**Genotyping and dataset preparation.** After shotgun screening, we selected the samples for deeper sequencing based on their endogenous DNA content, selecting those with the highest percent endogenous DNA and the location and period of the sampling sites to ensure an even spread. These shotgun sequencing reads from 39 ancient and modern black rats were cleaned and mapped to the de novo black rat genome assembly using the EAGER pipeline 1.92.55[110]. Within the pipeline, the adapters were removed with AdapterRemoval 2.2.0[111], reads were mapped with BWA 0.7.12 aln/samse algorithm[112], duplications were removed by DeDup 0.12.1 (https://github.com/apeltzer/DeDup) and damage patterns of each library were checked with mapDamage 2.0.6[113]. For the seven ds_halfUDG libraries, we masked 2 bp from both ends of the reads using trimBam in bamUtil 1.0.13[114] to remove the damaged sites.

The shotgun sequencing reads from four modern individuals, including the Californian black rat for de novo genome assembly, two published brown rat individuals (Accession: ERS215789, ERS215791)[67] and one published Asian house rat individual (Accession: SRS1581480, HXM4)[115] were mapped to the genome assembly using the BWA 0.7.12 mem algorithm. After using a mapping quality filtering of 30 and removing reads with multiple hits, duplications were removed using DeDup. We then performed indel realignment for cleaned bam files of both ancient and modern individuals using RealignerTargetCreator and IndelRealigner in The Genome Analysis Toolkit (GATK) v3.5-0[116].

For the demographic history analysis, we called diploid genotypes from three modern genomes using the highest coverage genome of each of the species: black rat (CP-5999), brown rat (ERS215791) and Asian house rat (HXM4). Each of the bam files was piled up using samtools mpileup, using reads with mapping quality and base quality over 30, and BAQ disabled. Bi-allelic SNPs were then individually called using bcftools call -m mode and filtered for SNPs with phred-scaled quality score (QUAL) over 30, sequence depth between 0.5-2X mean coverage, and not within 5 bp of an indel. After masking for repetitive regions, the consensus sequences of 18 largest autosomal scaffolds were generated, with heterozygous sites represented by IUPAC nucleotide code.

We called the pseudo-haploid genotypes in autosomal regions, from all modern and ancient individuals using ANGSD 0.931[117], with parameter "-doHaploCall 1" to randomly sample one base. As the 18 longest autosomal scaffolds covered >99%

of the autosomal assembly, we only called genotypes on the non-repetitive regions of these 18 scaffolds. We applied "-remove_bads 1 -uniqueOnly 1 -minMapQ 30 -minQ 30 -C 50 -baq 1" parameters to filter out reads that had multiple hits, with mapping quality or base quality less than 30, perform base alignment quality (BAQ) computation and adjust mapping quality based for excessive mismatches[118]. To remove the deamination-induced damages in ancient DNA molecules, we only kept the transversion variants for downstream analysis. The genotypes on single-copied male-specific Y-chromosome regions (scpMSY) were called from all male individuals using ANGSD 0.931, with the same filters as autosomal genotyping, and -doHaploCall 2 to get the major call. The detailed information of scpMSY regions identification was provided in Supplementary Note 1.

To estimate the heterozygosity rates of ancient rat samples, the cleaned reads with base quality and mapping quality over 30 were piled up with mpileup in SAMtools 1.3[119]. We then called pseudo-diploid genotypes with pileupCaller 1.2.2 (https://github.com/stschiff/sequenceTools) under random diploid calling mode, which randomly sampled two reads at each site, on the transversion variants identified in ANGSD. The heterozygosity rates calculated from pseudo-diploid genotypes were half of the real heterozygosity rates of the samples on these variants.

The sequencing reads of all the screened black rats after AdapterRemoval were also mapped to black rat reference mitochondrial sequence NC_012374.1 with BWA 0.7.12 aln/samse algorithm and realigned with CircularMapper 1.0[110]. The reads of brown rat and Asian house rat individuals were mapped to mitochondrial references of the brown rat (NC_001665.2) and the Asian house rat (NC_011638.1), respectively. After removing duplication using DeDup, the consensus sequences were generated by Schmutzi with a quality threshold of 30[120]. All the samples with <6000 missing sites in the consensus sequences were included in the downstream mitogenome analysis.

**Demographic history analysis.** The population size dynamics was estimated using PSMC 0.6.5[66], with parameter "-N25 -t20 -r5 -p "4 + 25*2 + 4 + 6"" and 100 bootstrap replicates. The PSMC output was visualised with generation time of 0.5 years and mutation rate $\mu = 2.96*10^{-9}$ site/generation, based on an estimate calculated in a previous study of the brown rat[67].

G-PhoCS 1.2.3[68] was applied to estimate the population sizes, population divergence times and migration rates among three rat species, using the three high-coverage diploid genomes. The analysis was performed on 38,078 loci of 1 kb length, identified in non-repetitive, autosomal regions. A preliminary analysis with all possible migration events was first run for 250,000 generations, then two parallel runs for 500,000 generations with one migration event were carried out for parameter estimation. Finally, the estimated parameters were converted to effective population sizes (Ne), divergence times (T) and total migration rates (m_total) as described in[68]: theta = 4*Ne*μ, tau = T*μ/g and m_total = m*tau, with mutation rate $\mu = 2.96*10^{-9}$ site/generation and generation time (g) of 0.5 years. The detailed information for loci selection and analysis was provided in Supplementary Note 2.

**Phylogenetic analysis.** The 67 ancient and 3 modern newly assembled mitochondrial genomes were analysed alongside seven modern reference genomes, including the modern Californian black rat from the reference genome assembly, two published brown rat individuals[67], one published Asian house rat individual[115] and the published mitochondrial genome references of the three species (black rat NC_012374.1, Asian house rat NC_011638.1, brown rat NC_001665.2). The haplotypes were aligned using MUSCLE v3.8.1551[121] with default parameters. Overall, 47 mitochondrial sequences from 77 individuals were involved in the analysis. The best-fit model was selected based on Akaike Information Criterion (AIC) calculated by jmodeltest v2.1.10 [122]. The Maximum Likelihood (ML) tree was built using RAxML v8.2.12[123], with GTR + I + G model and 100 bootstrap replicates.

The cytb region of the mitochondrial genome haplotypes were extracted using MEGA7, and combined with modern cytb haplotypes from previous publications[10,24,69,70] and this study. We aligned the data using MAFFT v7.123b[124], then built a ML tree using RAxML v8.2.9[123], with GTR + I + G model and 1000 bootstrap replicates.

The autosomal phylogeny was reconstructed using the neighbour-joining (NJ) method implemented in package Ape 5.3 in R 3.5.1, for 43 individuals, including 36 ancient black rats, 4 modern black rats, two brown rats, and one Asian house rat. The distance matrix was calculated based on 3,393,710 autosomal transversion variants, after removing singletons, using the genetic distance described in Gronau et al. [68]. Bootstrapping was performed by resampling the variants from 100 kb non-overlapping windows, and the support on each node was calculated based on 100 bootstrap replicates. The phylogenetic tree based on Y-chromosome scpMSY regions was built with RaxML 8.2.12[123], using GTR substitution model, ML estimation of base frequencies and 100 rapid bootstrapping replicates.

**Population genetics analysis.** The IBS distance matrix among individuals was calculated using PLINK v1.90b[125] with parameter "-distance 1-ibs". MDS analysis was performed using PLINK and ten dimensions were calculated on both datasets including all studied individuals and black rat individuals only. The $f_4$-statistics

were calculated by *qpDstat* 755 in ADMIXTOOLS 5.1 package[126], with parameter "f4 mode: YES", and the two brown rat individuals were used as the outgroup in all the analyses. The standard errors (SE) of $f_4$-statistics were estimated using jackknife among autosomal scaffolds.

We also applied Treemix 1.13[84] to simultaneously infer the population structure and admixture events among black rat populations. The black rat samples were grouped based on the geographic location, time period and phylogenetic pattern identified in previous analysis (Supplementary Data 2). The allele frequency was calculated by PLINK and 1,145,713 sites covered in at least one sample from each group were included in the analysis. We built the admixture graph assuming 0–10 migration events, with parameters "-k 500 -global -se -noss -root tanezumi" to group 500 SNPs per block for covariance matrix estimation. We then performed a global rearrangement after adding all the populations, calculated standard errors of migration weights, disabled sample size correction and assigned the Asian house rat as the root of the topology.

**Reporting summary**. Further information on research design is available in the Nature Research Reporting Summary linked to this article.

## Data availability

The black rat genome assembly is available in the NCBI under the accession number GCA_011800105.1. Aligned reads from the 39 newly reported ancient and modern black rats are available at the ENA archive under the accession number PRJEB47337. The mitochondrial haplotypes are available at NCBI GenBank under the accession number OK210796–OK210933 for CYTB regions and OM574930–OM574970 for mitochondrial genomes. Source data are provided as a Source Data file, with Supplementary Data 1 and 4.

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

## Acknowledgements

We thank the wet laboratory teams at MPI-SHH, the PalaeoBARN at the University of Oxford and BioArch at the University of York. We thank David K. James and Lucia Hui of the Alameda County Vector Control Services District for procuring the rat used for the de novo genome. We are grateful to Sarah Nagel at Max Planck Institute for Evolutionary Anthropology for the single-stranded library preparation, and Dovetail Genomics for the de novo genome assembly service. We thank Maria Spyrou for her suggestions and comments. We acknowledge Ewan Chipping and Helena England (University of York), Carl Phillips, Veronica Lindholm (Ålands Museum), Christine McDonnell and Nienke van Doorn (York Archaeological Trust), Emile Mittendorf (Gemeente Deventer), Inge Riemersma (Archaeological depot, Provincie Zuid-Holland), the Turkish Ministry of Culture & Tourism and the Çanakkale Museum, Jan Frolík and Iva Herichová (Institute of Archaeology of the Czech Academy of Sciences, Prague), Franz Humer and Eduard Pollhammer (Archaeological Park Carnuntum), Dorottya B. Nyékhelyi and László Daróczi-Szabó (Budapest History Museum), Institut National du Patrimoine (Tunisia), University of Barcelona, Spanish Ministry of Science and Innovation (Project HUM2006-03432/HIST), Spanish Ministry of Culture (programme of archaeological excavations abroad 2009); Spanish Agency of International Cooperation for the Development (2009), Catalan Institute of Classical Archaeology (ICAC), Vila Franca Municipal Museum and N'Zinga Oliveira (University of the Azores), Vujadin Ivanisević and Ivan Bugarski (Institute of Archaeology, Belgrade), Martin Nadler, M.A. and Dr. Silvia Codreanu-Windauer (Nürnberg and Regensburg respectively, Bavarian State Office for the Preservation of Monuments), the Field Museum of Natural History Chicago, The Natural History Museum London, and the American Museum of Natural History for providing materials and support. Funding was provided by the European Research Council (ERC-StG-337574-UNDEAD to G.L.; ERC-StG-206148 to N.B.; ERC-StG-716298 to S.V.-L.), the Natural Environment Research Council Doctoral Training Programme (A.J.), Wellcome (Small Grant in Humanities and Social Science 209817/Z to D.O.), the British Academy/Leverhulme Trust (SG170938 to D.O.), Estonian Research Council (PRG29 to E.R.), Czech Academy of Sciences (RVO:67985912 to R.K.), and Leverhulme Trust (ECF-2017-315 to A.H.B.), National Science Foundation (BCS 1123091 to J.F./S.W.-J.), Arts and Humanities Research Council (AH/J502716/1 to S.W.-J./J.F.). The de novo genome assembly, population genomics study, and radiocarbon dating were funded by the Max Planck Society.

## Author contributions

J.K., G.L. and D.O. designed the project; A.J., B.K.-K., B.K., C.S., H.A.-J., H.E., A.T. generated data; H.Y., A.J. and D.O. analysed data; G.A, N.B., H.B., B.B.-A., W.B., A.C., C.J.C., T.C., K.D., K.E., J.F., L.G., E.G., P.G., S.G., S.H.-D., C.H., R.H., N.K., H.K., Z.K., K.K., R.K., A.L., B.M.-T., N.M., A.M.-M., M.N., T.O'C., T.Ou, E.Q.M., K.P., J.P., N.P., S.R., J.R., E.R., J.B.S., J.S.G., E.T., S.V.-L., I.van der J., W.V.N., J.-D.V., T.W., S.W.-J. and J.Z. provided material and support; H.Y., D.O., A.J., A.H.-B., G.L. and J.K. wrote the paper with contributions from all other authors.

## Funding

## Competing interests

The authors declare no competing interests.

## Additional information

[1]Department of Archaeogenetics, Max Planck Institute for Evolutionary Anthropology, 04103 Leipzig, Germany. [2]School of Life Sciences, Peking University, 100871 Beijing, China. [3]Department of Archaeogenetics, Max Planck Institute for the Science of Human History, 07745 Jena, Germany. [4]Palaeogenomics & Bio-Archaeology Research Network, Research Laboratory for Archaeology and History of Art, University of Oxford, Oxford OX1 3QY, UK. [5]Department of Archaeology, Classics and Egyptology, University of Liverpool, Liverpool L69 7WZ, UK. [6]Research Centre in Evolutionary Anthropology and Palaeoecology, Liverpool John Moores University, Liverpool L3 3AF, UK. [7]Museum of Vertebrate Zoology, University of California, Berkeley, Berkeley, CA 94720-3160, USA. [8]Department of Archaeology, University of York, York YO1 7EP, UK. [9]BioArCh, Department of Archaeology, University of York, York YO1 7EP, UK. [10]Department of Anthropology, University of British Columbia, Vancouver, BC, Canada. [11]Department of Ecology and Evolutionary Biology, Cornell University, Ithaca, NY 14853, USA. [12]Charité – Universitätsmedizin Berlin, Institut für Pathologie, Charitéplatz 1, 10117 Berlin, Germany. [13]Postgraduate Institute of Archaeology, 407, Bauddhaloka Mawatha, Colombo 7, Sri Lanka. [14]Römisch-Germanisches Zentralmuseum, Leibniz-Forschungsinstitut für Archäologie, Ernst-Ludwig-Platz 2, 55116 Mainz, Germany. [15]Christian Archaeology and Byzantine Art History, Philipps University of Marburg, 35037 Marburg, Germany. [16]Department of History and Archaeology, University of Ruhuna, Matara 81000, Sri Lanka. [17]Department of Archaeology,

Max Planck Institute for the Science of Human History, 07745 Jena, Germany. [18]School of Social Science, The University of Queensland, St Lucia, QLD, Australia. [19]Archaeozoology, Archaeobotany, Societies, Practices, Environments (AASPE-UMR7209), CNRS, National Museum of Natural History (MNHN), Paris, France. [20]Archeoplan Eco, 2616 LZ Delft, Netherlands. [21]Department of Anthropology, Rice University, 6100 Main St, Houston, TX 77005, USA. [22]Archaeological Services, University of Durham, Durham, UK. [23]Ukrainian Scientific Center of Ecology of the Sea, Odessa 65009, Ukraine. [24]Schmalhausen Institute of Zoology, National Academy of Sciences of Ukraine, Kyiv 01030, Ukraine. [25]Negaunee Integrative Research Center, Field Museum of Natural History, Chicago, IL 60605, USA. [26]Department of Archaeology & Anthropology, Bournemouth University (Visiting Fellow), Poole BH12 5BB, UK. [27]Canterbury Archaeological Trust, 92a Broad Street, Canterbury, Kent CT1 2LU, UK. [28]6 Fell View Park, Gosforth, Seascale, Cumbria CA20 1HY, UK. [29]L'Ecole Tunisienne de l'Histoire et l'Anthropologie, Tunis, Tunisia. [30]University of Tunis, Tunis, Tunisia. [31]Department of Culture, University of Helsinki, P.O. Box 59, FI-00014 Helsinki, Finland. [32]Osteological Research Laboratory, University of Stockholm, 10691 Stockholm, Sweden. [33]Freelance archaeozoologist, Liliom u. 4. 1/1, Balatonfüred 8230, Hungary. [34]VIAS Vienna Institute for Archaeological Science, University of Vienna, Althanstraße 14, 1090 Vienna, Austria. [35]Department of Natural Sciences and Archaeometry, Institute of Archaeology of the Czech Academy of Sciences, Prague, Letenská 4, 118 01, Prague, Czech Republic. [36]Centre for Palaeogenetics & Department of Geological Sciences, Stockholm University, Stockholm 10691, Sweden. [37]Institut National de Patrimoine, Tunis 1008, Tunisia. [38]Institute of Archaeology, 11000 Belgrade, Serbia. [39]Departmento de Biología, Universidad Autónoma de Madrid, 28049 Madrid, Spain. [40]Institute of Archaeology, University College London, London WC1H 0PY, UK. [41]Centro de Arqueologia da Universidade de Lisboa (UNIARQ), Faculdade de Letras, Universidade de Lisboa, 1600-214 Lisboa, Portugal. [42]Centre National de la Recherche Scientifique, University of Lille, Lille, France. [43]Department of Anthropology, University of California, Santa Cruz, 1156 High St, Santa Cruz, CA 95064, USA. [44]Department of Philosophy, Institute of Prehistoric Archaeology, Friedrich-Alexander-University of Erlangen-Nürnberg, 91054 Erlangen, Germany. [45]Department of Archaeology, Sir Marcus Fernando Mawatha, Colombo 07, Sri Lanka. [46]Austrian Academy of Sciences, Austrian Archaeological Institute, Hollandstraße 11-13, 1020 Vienna, Austria. [47]Consell Insular d'Eivissa i Formentera, Avenida de España 49, 07800 Eivissa, Illes Balears, Spain. [48]Department of Archaeology, Institute of History and Archaeology, University of Tartu, 2 Jakobi St, 51005 Tartu, Estonia. [49]Secció de Prehistòria i Arqueologia, University of Barcelona, Barcelona, Spain. [50]Department of Archaeology, Durham University, Durham, UK. [51]Archaeology of Social Dynamics, IMF-CSIC, Barcelona 08001, Spain. [52]Cultural Heritage Agency of the Netherlands, Smallepad 5, 3811 MG Amersfoort, The Netherlands. [53]Royal Belgian Institute of Natural Sciences, Vautierstraat 29, 1000 Brussels, Belgium. [54]Laboratory of Biodiversity and Evolutionary Genomics, Katholieke Universiteit Leuven, 3000 Leuven, Belgium. [55]Department of Archaeology, University of Reading, Berkshire RG6 6AB, UK. [56]ArchaeoBone, Blekenweg 61, 9753 JN Haren, The Netherlands. [57]Department of Archaeology, University of Sydney, Sydney, NSW 2006, Australia. [58]Department of Archaeology, University of Aberdeen, Aberdeen AB24 3UF, UK. [59]Department of Archaeology, Simon Fraser University, Burnaby, BC V5 1S6, Canada. [60]Institute of Clinical Molecular Biology, Kiel University, Kiel 24105, Germany. [61]These authors contributed equally: He Yu, Alexandra Jamieson. ✉email: krause@eva.mpg.de; greger.larson@arch.ox.ac.uk; david.orton@york.ac.uk

