## [Peer Review File · Nature Communications]

Palaeogenomic analysis of black rat (*Rattus rattus*) reveals multiple European introductions associated with human economic historyREVIEWER COMMENTS

Reviewer #1 (Remarks to the Author):

The study by Yu et al. presents an extensive and impressive dataset of ancient and modern black rat genomes spanning over 15 centuries and a wide geographical range of samples, with the main aim to understand and investigate the dispersal of black rats into the Mediterranean and Europe and its relationship with Europe's population history from the 1st to 17th centuries.

The study is interesting and falls within the scope of Nature Communications. Lab procedures meet the required standard for ancient DNA analysis, and the results and conclusions are supported by the analyses they present. The manuscript is well written, but needs some improvement as there are several aspects and sections that need correction and clarification. So I would like to ask the authors to add some additional information for better understanding. I am outlining more specific comments in the following lines:

MAIN TEXT

Line 218 of the main text mentions Supplementary Figures 11 and 12, however Supp. Figure 11 corresponds to Treemix admixture graphs and matrices and there is no Supp. Figure 12. Please fix accordingly.

Line 233 - a mutation rate of 2.96×10^{-9} per generation is stated. Please be consistent when mentioning the mutation rate both in the methods section and figures.

So you have 199 DNA extracts combining both the 191 ancient and the 8 modern ones from Zembra. However there are a couple of things that are confusing and require clarification:

1. Why do you only have 81 Sample IDs in Supp. Table 7? It is unclear if you actually sampled 199 individuals or produced 199 aDNA extracts. Please explain.
2. In line 274 of the main text you say you sampled 191 ancient rat individuals from 33 archaeological sites, but in the corresponding supplementary Note 1, you mention 200 ancient rat individuals from 34 archaeological sites. Which one is correct then?

Please explain and correct/complete the corresponding tables as needed and clarify this on the main text and supplementary accordingly.

Also, as the 8 modern individuals from Tunisia are included in Supp. Table 7, it would be less confusing if you move the reference to the table to the end of the sentence (line 276).

In line 277 of main text you say you retrieved 70 ancient mitochondrial genomes, but in the corresponding Supp. Table 8 you have 74 sample IDs. Is this a mistake? Did you discard 4 samples? Please explain and fix accordingly. Also, in the table there is information missing for 4 samples, please complete or explain why the information is not available for those samples.

Even though you have samples with great endogenous content (Supp. Table 7), they are not all included/considered. Could you please explain and elaborate on the inclusion parameters to decide which samples to use for each of the analyses?

Also, did the variable mtDNA coverage between the samples (3.5x-300.0x) affect the calling of haplotypes? How many ancient samples does the major clade include? Please add this information to the main text.

In line 289 of main text, what is the total number of sequences used for the cytochrome b analysis? As I understand you used your 70 ancient mitochondrial genomes, plus the 132 previously unpublished modern CYTB sequences from across the Indian Ocean basin, and how many other sequences published in previous studies?

In the main text it is not really clear that the 36 shotgun sequenced ancient individuals for whole

genome analysis are a subset of the 70 from the mtDNA analysis (line 348). How and why did you end up choosing only 36? It would be good to discuss and describe a little about the authenticity of the ancient data - endogenous DNA content, C-T deamination damage levels, contamination levels, etc. - and include the corresponding references to the Supp. Tables. Please elaborate and include this information in the main text if possible.

I'm wondering why the authors chose not to do any of the more common exploratory analysis for population structure based on autosomal SNPs such as PCA or an unsupervised Admixture analysis which could also help answer the question of population continuity. I would expect this type of analysis to confirm the results they are presenting but it would be very interesting to see how they compare to each other (PCA vs IBS-MDS). Please elaborate on this and include such analysis if possible.

A question more out of curiosity: why do the authors choose to plot and show a visualisation of the f4 values instead of the z-scores in Figure 4?

In line 497 it is suggested that the inferred relict population may come from an unsampled region of France or southwest Europe. Could you please elaborate more on why southern France and not Italy or even the Iberian Peninsula, could be the place of origin or persistence of the black rat population?

In line 518 please replace "suggest" with "confirm" as used previously in both the abstract and the results and discussion section.

In line 645, apart from the reference, please add a brief description of the method used for the extractions at the University of York as you do for the other two institutions.

In line 665, is the Danish National High-Throughput Sequencing Centre for screening really in Sacramento and part of Novogene? Sounds a little odd. If not, please correct. Also, where else where the rest of the extracts - the ones from Jena - sequenced? Please add the info.

Typo in line 690.

SUPPLEMENTARY NOTES AND TABLES

In general the order and organisation of the Supplementary Notes and Figures is quite confusing and makes it hard to navigate and find information in the Supp. document. Please modify the order to be in agreement with the appearance and mention in the main text. Also, please reference each of the sections in main text accordingly - there are some missing.

Some of the Supplementary Tables are missing information in some fields. For example, Supp. Table 9 is missing location information for some of the samples. Is this information not available? If so, please state that somehow (i.e. NA). Also Supp. Table 19 is missing the information for Cyt b primer pair U4/L4. Please double check all tables and complete when possible.

The first paragraph of page 34 in the Supplementary is hard to understand and Supplementary Figure 6 is missing. Please modify in order to clarify and include the missing figure. Also add the corresponding mention to this figure, and all other supplementary figures, in the main text.

Please add title and descriptive legends to each of the Supplementary Tables.

Reviewer #2 (Remarks to the Author):

This manuscript discusses a large dataset of unpublished ancient mitogenomes, cytochrome B mitochondrial data and nuclear genomes from black rats. To facilitate their nuclear analysis, the authors also generated a de novo assembly genome of the black rat. Furthermore, to support their

inferences, they also utilized newly generated and published modern mitochondrial and nuclear data. The number of data generated and analyzed in this study is really impressive.

The authors used this data to:

- Evaluate phylogeographical patterns using mitogenomes, as well as cytochrome B mitochondrial sequences. This allowed them to further define mitochondrial clade sub-/structure and identify possible routes for the entrance of black rat mitochondrial lineages into Europe.
- Further explore population structure using nuclear data. The authors identified a population turnover in Europe between the 6th and 10th centuries CE.
- Used the de novo assembly to explore and compare demographic pattern among the black rat and the Asian and brown rats.

The manuscript is well written and easy to follow. The presentation items are also clear and help to follow the story line. The study is of significance for the field and presents novel results. The methodology used follows the standards for ancient DNA work, and the authors used a suite of standard analysis for this kind of work. I would like to recommend this manuscript for publication in Nature Communications.

However, I do have some comments and suggestions for the authors to be addressed. I would also like to mention that I am not familiar with the G-PhoCS analysis, so I have not reviewed that section. I hope another reviewer can comment on it.

In general, I suggest the authors to be more specific about the dataset size and composition used for the different analysis. Below, I have made some suggestions of places where this could be improved.

Minor comments and corrections:

- There is an inconsistent used of the Oxford comma throughout the manuscript. Some paragraphs used it, while in other is not used at all. For consistence, I suggest you correct this.
- Line 102. I suggest you mention that the ancient data was combined and analyzed together with published and newly generated modern data.
- I suggest that after defining the species common names in lines 111-112, the authors use them throughout the text. Specially, because the figures show the common name. Currently, there is a mix of common and scientific name, which reduces the reading flow.
- Line 171: "reappear" to past tense for consistency with the rest of the introduction
- Line 200. The abstract only mentioned the 67 ancient mitogenomes. I suggest rephrasing this sentence to "70 new mitochondrial genomes (ancient = 67 and modern = 3). Alternatively, modify the abstract to also mention the modern data included in the study.
- Line 202. Spell out mitochondrial cytochrome B (CYTB) as "mtDNA" can be misleading. I would also suggest to not only indicate 132 sequences, but indicate X = ancient, N = modern, C = published modern sequences.
- Line 203. In the abstract only the 36 ancient nuclear genomes are mentioned, I would suggest also to indicate how many modern and how many ancient were used in the analysis.
- Line 277, same as suggested for Line 200.
- Line 286 – 289. I suggest you indicate the full dataset analysed for the CYTB. Total number of samples (354), modern, ancient, published and unpublished.

- Line 326 Late Pleistocene with capital letter.

- Line 349 I would suggest you mention that the nuclear genome from the California sample was also included in some of the analysis (e.g., Figure 3). As their number of black rats used there is 40, not 39.

- General comment for the section “Radiocarbon dating and calibration”. When reporting stable isotope data, the delta symbol should be reported in italics.

<https://www.tandfonline.com/doi/pdf/10.1080/10256016.2011.645702>

Change (‰ vs. AIR) to (‰, AIR) and (‰ vs. VPDB) to (‰, VPDB).

- Line 661- 663. It is unclear which enzyme and conditions were used for the index PCR of the Oxford and York libraries. Could you specify it for reproducibility?

- Line 664 – 665. Did you screen the libraries at the Danish National High-Throughput Sequencing Centre? What about Novogene? Please, clarify this.

- Line 690. Missing a letter. It should be “We then...”

- Line 746. General comment for the section “Phylogenetic analysis”. As suggested for previous sections, please indicate the number of samples used for each of the analysis, age (ancient or modern) and whether it was new or published data.

- Line 788. This should be changed to “Aligned reads from the 36 newly reported ancient and three modern black...” or “Aligned reads from the 39 newly reported ancient and modern black...”

Other comments and suggestions:

- Lines 703 – 708. I would suggest moving the processing of the mitogenome reads to right after the paragraph about mapping to the nuclear genome (Lines 685 – 692) or to the bottom of this section. It would be easier for the reader to have all the nuclear processing together and then the mitochondrial.

- Did you try using BEAST to estimate the divergence of the mitochondrial lineages? Priors could be used to include uncertainty for the sample ages. A molecular dated phylogeny would add value to the mitogenome analysis, specially to explore the lineage turnover. Furthermore, as BEAST does not require an outgroup, it might increase the resolution of the mitogenome phylogeny. In the discussion, for instance line 528, changes in population size are also mentioned. Perhaps BEAST could have been used to estimate changes in population size. In my opinion this could have been explored a bit more.

Figures:

Comments for Figure 2:

- Legend Figure 2A. I would suggest using a comma to separate the location names and including the number of samples per location in the parenthesis.

- I am a bit confused by the number of circles (modern samples) in the panel A (map). I apologize if I am not interpreting it correctly. ZMB represents three samples, for which you have nuclear and mitochondrial data. What about the other modern dots without “archeological site initials”? Could this be clarified in the legend?

- Legend Figure 2B. Please, include total number of sequences, ancient sequences and modern published/unpublished.

Comments for Figure 3:

- Figure 3B. The “Northern group” label would be better at the top, close to the B, so it is matching the style used for the “Southern group”.

- In the Roman group, squares are used to represent two different sites. Could this be changed?

Point-by-point response

Reviewer #1 (Remarks to the Author):

The study by Yu et al. presents an extensive and impressive dataset of ancient and modern black rat genomes spanning over 15 centuries and a wide geographical range of samples, with the main aim to understand and investigate the dispersal of black rats into the Mediterranean and Europe and its relationship with Europe's population history from the 1st to 17th centuries.

The study is interesting and falls within the scope of Nature Communications. Lab procedures meet the required standard for ancient DNA analysis, and the results and conclusions are supported by the analyses they present. The manuscript is well written, but needs some improvement as there are several aspects and sections that need correction and clarification. So I would like to ask the authors to add some additional information for better understanding. I am outlining more specific comments in the following lines:

MAIN TEXT

Line 218 of the main text mentions Supplementary Figures 11 and 12, however Supp. Figure 11 corresponds to Treemix admixture graphs and matrices and there is no Supp. Figure 12. Please fix accordingly.

The figures related here are Supplementary Figures 1 and 2. We have fixed this in the main text.

Line 233 - a mutation rate of 2.96×10^{-9} per generation is stated. Please be consistent when mentioning the mutation rate both in the methods section and figures.

Thank you for pointing this out. We have corrected the figure and method section accordingly.

So you have 199 DNA extracts combining both the 191 ancient and the 8 modern ones from Zembra. However there are a couple of things that are confusing and require clarification:

1. Why do you only have 81 Sample IDs in Supp. Table 7? It is unclear if you actually sampled 199 individuals or produced 199 aDNA extracts. Please explain.
2. In line 274 of the main text you say you sampled 191 ancient rat individuals from 33 archaeological sites, but in the corresponding supplementary Note 1, you mention 200 ancient rat individuals from 34 archaeological sites. Which one is correct then?

Please explain and correct/complete the corresponding tables as needed and clarify this on the main text and supplementary accordingly.

Sorry for causing the confusion. The “Sample ID” in Sup Table 7 is the additional ID used in Jena for next generation sequencing. We sampled 199 individuals and generated one extract for each individual, with each extract ID in this table representing one individual. We have changed this column to “NGS ID” to avoid confusion.

The numbers listed in line 274 are correct. We have corrected in supplementary Note 1 to “199 ancient and modern rat individuals from 33 archaeological sites and one modern site”.

Also, as the 8 modern individuals from Tunisia are included in Supp. Table 7, it would be less confusing if you move the reference to the table to the end of the sentence (line 276).

We've moved the reference accordingly.

In line 277 of main text you say you retrieved 70 ancient mitochondrial genomes, but in the corresponding Supp. Table 8 you have 74 sample IDs. Is this a mistake? Did you discard 4 samples? Please explain and fix accordingly. Also, in the table there is information missing for 4 samples, please complete or explain why is the information not available for those samples.

In Sup Table 8, the four additional samples include one modern black rat from the USA, which we generated the *de novo* genome assembly from, and three modern brown rat and Asian house rat from previous publications. They are also the samples without site information. Now I have added the site information of the Californian modern rat as "California", and the three published individuals as "published".

Even though you have samples with great endogenous content (Supp. Table 7), they are not all included/considered. Could you please explain and elaborate on the inclusion parameters to decide which samples to use for each of the analysis?

After shotgun screening, we selected the samples for deeper sequencing based on their endogenous DNA and the sampling sites. For the sampling sites with a lot of high-quality samples, we only selected at most four samples per site, to avoid an over-representation of any site in our final dataset. After deeper sequencing, all the 39 ancient and modern samples were included in the nuclear genome analysis, with at least 0.1X coverage.

All the screening and deeper sequencing reads were mapped to the mitochondrial genome and samples with at most 6000 missing sites after haplotype calling were included in the mitogenome analysis. We've added the criteria in the method section.

Also, did the variable mtDNA coverage between the samples (3.5x-300.0x) affect the calling of haplotypes? How many ancient samples does the major clade include? Please add this information to the main text.

The coverage would affect the completeness of our haplotype calling. From Supplementary Table 8, we can find that the low-coverage samples have more missing sites, while after the coverage reaches 10x or more, the missingness is generally less than 1% and could be omitted. The major clade includes 48 samples, including 44 ancient black rats, 3 modern rats from Tunisia and one modern rat from California. I've added this information in the main text.

In line 289 of main text, what is the total number of sequences used for the cytochrome b analysis? As I understand you used your 70 ancient mitochondrial genomes, plus the 132 previously unpublished modern CYTB sequences from across the Indian Ocean basin, and how many other sequences published in previous studies?

There are in total 476 samples analyzed for cytochrome b, including our 70 mitochondrial genomes, 132 previous unpublished modern sequences, and 274 published sequences. We've added the number of published sequences in the main text.

In the main text it is not really clear that the 36 shotgun sequenced ancient individuals for whole genome analysis are a subset of the 70 from the mtDNA analysis (line 348). How and why did you

end up choosing only 36? It would be good to discuss and describe a little about the authenticity of the ancient data - endogenous DNA content, C-T deamination damage levels, contamination levels, etc. - and include the corresponding references to the Supp. Tables. Please elaborate and include this information in the main text if possible.

As described in the previous answer, we chose a maximum of four samples per site after shotgun screening, for whole genome deeper sequencing, and ended up with 39 nuclear genomes available for analysis (36 ancient and 3 modern individuals). But all the screened samples were used for mitochondrial haplotype reconstruction which gave us 70 samples with at least 2/3 sites covered. We've added the selection criteria in the method session.

I'm wondering why the authors chose not to do any of the more common exploratory analysis for population structure based on autosomal SNPs such as PCA or an unsupervised Admixture analysis which could also help answer the question of population continuity. I would expect this type of analysis to confirm the results they are presenting but it would be very interesting to see how they compare to each other (PCA vs IBS-MDS). Please elaborate on this and include such analysis if possible.

Thank you for your valuable suggestion. We also considered using PCA or Admixture on autosomal analysis. However, applications of these methods in ancient DNA studies (especially ancient human studies) rely much on a high-quality modern reference panel for calculating PCs or performing a stable population clustering. In our study, we unfortunately don't have such a reference panel for the black rat, and our ancient individuals varied in their nuclear genome coverage. This largely affected the statistical power of both of these methods. An advantage of IBS-MDS over PCA or Admixture is that it is based on pairwise genetic distance, instead of an overall comparison among all individuals, which is less affected by the low coverage samples. That's why we decided to include IBS-MDS in our manuscript.

We are also including our trial PCA result with smartpca here, including only black rat individuals. We found that each PC only explained for a small proportion of variation among the rats, and the first two PCs were dominated by several highly diverse southern groups. When we removed these outlier groups, a similar Roman/Byzantine vs. Medieval distinction can also be observed, kind of similar to what we observed in IBS-MDS analysis.

A question more out of curiosity: why do the authors choose to plot and show a visualisation of the f_4 values instead of the z-scores in Figure 4?

Because the f_4 values reflect the true value of the statistics, while the Z-scores corresponding to standard errors would be impacted by the usable SNP numbers in the test. As our ancient samples varied in their coverages and available SNP numbers, we think the f_4 values with standard error provide a more reasonable comparison among individuals.

In line 497 it is suggested that the inferred relict population may come from an unsampled region of France or southwest Europe. Could you please elaborate more on why southern France and not Italy or even the Iberian Peninsula, could be the place of origin or persistence of the black rat population?

The Iberian Peninsula is what we meant by ‘southwest Europe’ - we have replaced the latter with the former in the text. Italy is also a theoretically possible location for the inferred relict population, but we think this is less likely since (a) it is a likely a source for the second, genetically distinct wave of introduction (discussed in the following paragraphs of the main text), and (b) it would fit less well with the greater affinity between medieval and Roman rats seen in western than in central Europe.

In line 518 please replace "suggest" with "confirm" as used previously in both the abstract and the results and discussion section.

We have changed it accordingly.

In line 645, apart from the reference, please add a brief description of the method used for the extractions at the University of York as you do for the other two institutions.

We have added more details on the extraction method at York, as described for the other two institutions.

In line 665, is the Danish National High-Throughput Sequencing Centre for screening really in Sacramento and part of Novogene? Sounds a little odd. If not, please correct. Also, where else where the rest of the extracts - the ones from Jena - sequenced? Please add the info.

The screening was carried out at the Danish National High-Throughput Sequencing Centre. We’ve corrected in the method session accordingly. The extracts from Jena were sequenced in Jena as well.

Typo in line 690.

We have corrected the line in the main text.

SUPPLEMENTARY NOTES AND TABLES

In general the order and organisation of the Supplementary Notes and Figures is quite confusing and makes it hard to navigate and find information in the Supp. document. Please modify the order to be in agreement with the appearance and mention in the main text. Also, please reference each of the sections in main text accordingly - there are some missing.

We have reordered the supplementary sessions and added a table of contents in the first page. We also checked the main text, and altered the citations of supplementary contents accordingly.

Some of the Supplementary Tables are missing information in some fields. For example, Supp. Table 9 is missing location information for some of the samples. Is this information not available? If so, please state that somehow (i.e. NA). Also Supp. Table 19 is missing the information for Cyt b primer pair U4/L4. Please double check all tables and complete when possible.

Thank you for pointing this out. In Sup Table 9, we don't have the exact location information. We have therefore filled those fields with "NA".

The first paragraph of page 34 in the Supplementary is hard to understand and Supplementary Figure 6 is missing. Please modify in order to clarify and include the missing figure. Also add the corresponding mention to this figure, and all other supplementary figures, in the main text.

We apologize for the mistake. We decided to remove Supplementary Figure 6, as it duplicated with Sup Figure 4 in their content, but didn't change the Figure numbering accordingly. We have therefore removed this paragraph and shifted the numbering of supplementary figures.

Please add title and descriptive legends to each of the Supplementary Tables.

We have added the title and legends to each of the supplementary tables, including the ones presented in separate spreadsheets, now labelled "Supplementary Data".

Reviewer #2 (Remarks to the Author):

This manuscript discusses a large dataset of unpublished ancient mitogenomes, cytochrome B mitochondrial data and nuclear genomes from black rats. To facilitate their nuclear analysis, the authors also generated a de novo assembly genome of the black rat. Furthermore, to support their inferences, they also utilized newly generated and published modern mitochondrial and nuclear data. The number of data generated and analyzed in this study is really impressive.

The authors used this data to:

- Evaluate phylogeographical patterns using mitogenomes, as well as cytochrome B mitochondrial sequences. This allowed them to further define mitochondrial clade sub-/structure and identify possible routes for the entrance of black rat mitochondrial lineages into Europe.
- Further explore population structure using nuclear data. The authors identified a population turnover in Europe between the 6th and 10th centuries CE.
- Used the de novo assembly to explore and compare demographic pattern among the black rat and the Asian and brown rats.

The manuscript is well written and easy to follow. The presentation items are also clear and help to follow the storyline. The study is of significance for the field and presents novel results. The methodology used follows the standards for ancient DNA work, and the authors used a suite of standard analysis for this kind of work. I would like to recommend this manuscript for publication in Nature Communications.

However, I do have some comments and suggestions for the authors to be addressed. I would also like to mention that I am not familiar with the G-PhoCS analysis, so I have not reviewed that section. I hope another reviewer can comment on it.

In general, I suggest the authors to be more specific about the dataset size and composition used for the different analysis. Below, I have made some suggestions of places where this could be improved.

We thank the reviewer for the recognition of our manuscript, and we do realize that our multiple lines of analysis with different datasets would lead to some confusion. We reply to the comments and suggestions below.

Minor comments and corrections:

- There is an inconsistent use of the Oxford comma throughout the manuscript. Some paragraphs used it, while in others it is not used at all. For consistency, I suggest you correct this.

We have now used Oxford commas throughout.

- Line 102. I suggest you mention that the ancient data was combined and analyzed together with published and newly generated modern data.

We have mentioned this in the revised abstract.

- I suggest that after defining the species common names in lines 111-112, the authors use them throughout the text. Specially, because the figures show the common name. Currently, there is a mix of common and scientific name, which reduces the reading flow.

We have changed most of the scientific names in the text to common names.

- Line 171: “reappear” to past tense for consistency with the rest of the introduction

We have made this change.

- Line 200. The abstract only mentioned the 67 ancient mitogenomes. I suggest rephrasing this sentence to “70 new mitochondrial genomes (ancient = 67 and modern = 3). Alternatively, modify the abstract to also mention the modern data included in the study.

We have rephrased the introduction part by mentioning ancient and modern mitochondrial genomes separately.

- Line 202. Spell out mitochondrial cytochrome B (CYTB) as “mtDNA” can be misleading. I would also suggest to not only indicate 132 sequences, but indicate X = ancient, N = modern, C = published modern sequences.

We have changed “mtDNA” to “mitochondrial DNA” and indicated the number of museum and modern samples accordingly.

- Line 203. In the abstract only the 36 ancient nuclear genomes are mentioned, I would suggest also to indicate how many modern and how many ancient were used in the analysis.

As above, we also rephrased to mention ancient and modern genomes separately.

- Line 277, same as suggested for Line 200.

As above, we add “67 ancient and 3 modern” in that line.

- Line 286 – 289. I suggest you indicate the full dataset analysed for the CYTB. Total number of samples (354), modern, ancient, published and unpublished.

We've added the total number of samples, as well as the number of ancient / modern / unpublished / published samples in the main text.

- Line 326 Late Pleistocene with capital letter.

We made this correction.

- Line 349 I would suggest you mention that the nuclear genome from the California sample was also included in some of the analysis (e.g., Figure 3). As their number of black rats used there is 40, not 39.

We now describe in Line 357-358 of the main text about the California sample as well as the published brown rats and Asian house rat.

- General comment for the section “Radiocarbon dating and calibration”. When reporting stable isotope data, the delta symbol should be reported in italics.

<https://www.tandfonline.com/doi/pdf/10.1080/10256016.2011.645702>

Change (‰ vs. AIR) to (‰, AIR) and (‰ vs. VPDB) to (‰, VPDB).

We made these changes.

- Line 661- 663. It is unclear which enzyme and conditions were used for the index PCR of the Oxford and York libraries. Could you specify it for reproducibility?

We have added this information in the revised method section.

- Line 664 – 665. Did you screen the libraries at the Danish National High-Throughput Sequencing Centre? What about Novogene? Please, clarify this.

The screening was carried out at the Danish National High-Throughput Sequencing Centre and it is not related to Novogene. We have removed the Novogene reference in the main text.

- Line 690. Missing a letter. It should be “We then...”

We've corrected the text.

- Line 746. General comment for the section “Phylogenetic analysis”. As suggested for previous sections, please indicate the number of samples used for each of the analysis, age (ancient or modern) and whether it was new or published data.

We have added the numbers of individuals/haplotypes involved in each analysis, as well as their age/publication groups in the method section.

- Line 788. This should be changed to “Aligned reads from the 36 newly reported ancient and three modern black...” or “Aligned reads from the 39 newly reported ancient and modern black...”

We have changed this to “39 newly reported ancient and modern black rats”.

Other comments and suggestions:

- Lines 703 – 708. I would suggest moving the processing of the mitogenome reads to right after the paragraph about mapping to the nuclear genome (Lines 685 – 692) or to the bottom of this section. It would be easier for the reader to have all the nuclear processing together and then the mitochondrial.

We have moved the paragraph of mitogenome processing to the end of this section, to keep all the nuclear processing information together.

- Did you try using BEAST to estimate the divergence of the mitochondrial lineages? Priors could be used to include uncertainty for the sample ages. A molecular dated phylogeny would add value to the mitogenome analysis, specially to explore the lineage turnover. Furthermore, as BEAST does not require an outgroup, it might increase the resolution of the mitogenome phylogeny. In the discussion, for instance line 528, changes in population size are also mentioned. Perhaps BEAST could have been used to estimate changes in population size. In my opinion this could have been explored a bit more.

Thank you for this valuable suggestion. We didn't apply BEAST in the mitochondrial analysis for two reasons. First, our samples only span ~1,500 years, from Roman to late medieval time period, while the split among mitochondrial haplogroups is probably several times older than that. Second, the uncertainty of our sample dates could be over 200 years, which is quite large compared to an overall time span of 1,500 years.

When we try using BEAST on the mitogenome phylogeny analysis without outgroups (as figure below), we get a similar phylogeny like the Maximum Likelihood tree using RaxML, with two well-supported clades, and little structure within each clade. The split time of these two clades is estimated to be 12,700 BP, with a huge 95% HPD interval of 20,200-4,800 BP. The large estimated interval could be due to the relatively recent ages of our ancient samples, compared to the much older lineage split event.

We also reconstruct the Bayesian skyline plot based on these black rat mitogenomes, for estimating the changes in the rat population size (shown in figure below). The plot reveals a population decline and resurgence between 2000-1500 BP, but considering the large uncertainty of tip dating, which is revealed in the dated phylogenetic tree, this estimation of decline and resurgence timing lack precision.

For the above reasons, we decided not to include the BEAST analysis in our manuscript, as the result is unstable and does not tell us much more than what we were able to conclude from the other analyses.

Figures:

Comments for Figure 2:

- Legend Figure 2A. I would suggest using a comma to separate the location names and including the number of samples per location in the parenthesis.

We have added the number of samples involved in mitochondrial/nuclear analysis on each site in parentheses.

- I am a bit confused by the number of circles (modern samples) in the panel A (map). I apologize if I am not interpreting it correctly. ZMB represents three samples, for which you have nuclear and mitochondrial data. What about the other modern dots without “archeological site initials”? Could this be clarified in the legend?

The other modern dots without site names shown in panel A, are the sites from previous studies with mitochondrial data included in our analysis. We realise that including published sites in this panel would cause confusion, so in the revised figure we removed these dots.

- Legend Figure 2B. Please, include total number of sequences, ancient sequences and modern published/unpublished.

We have added the sample numbers in the legend, with a total of 476 samples, including 67 ancient ones, 135 modern ones and 274 published ones.

Comments for Figure 3:

- Figure 3B. The “Northern group” label would be better at the top, close to the B, so it is matching the style used for the “Southern group”.

We put the “Northern group” label next to the division line in the first submission to emphasize the split between the two groups. We also rotated both the labels so that they are now vertical which matches the style in Figure 3A.

- In the Roman group, squares are used to represent two different sites. Could this be changed?

We’ve changed the symbol to distinguish the two sites. We also changed the symbols in supplementary figures 7 and 8.

REVIEWER COMMENTS

Reviewer #2 (Remarks to the Author):

Dear authors,

Thank you for your efforts answering reviewers' questions and implementing all the changes to the texts, figures and supplementary material. These changes have improved the readability of the manuscript, making it more understandable for other researchers.

I agree with Reviewer #1 that the paper falls within the scope of Nature Communications, follows standards for ancient DNA work, and conclusions are supported by the analyses.